# Effect of a Fortified Biostimulant Extract on Tomato Plant Productivity, Physiology, and Growing Media Properties

**DOI:** 10.3390/plants13010004

**Published:** 2023-12-19

**Authors:** Marianne Weisser, Scott William Mattner, Liam Southam-Rogers, Graham Hepworth, Tony Arioli

**Affiliations:** 1Seasol R&D Department, Bayswater, VIC 3155, Australia; tonyarioli@seasol.com.au; 2VSICA (Victorian Strawberry Industry Certification Authority) Research, Toolangi, VIC 3777, Australia; swmattner@hotmail.com; 3School of Agriculture, Biomedicine and Environment, La Trobe University, Melbourne, VIC 3086, Australia; 4Applied Horticultural Research, Eveleigh, NSW 2015, Australia; liam@ahr.com.au; 5Statistical Consulting Centre, School of Mathematics and Statistics, The University of Melbourne, Parkville, VIC 3010, Australia; hepworth@unimelb.edu.au; 6School of Life & Environmental Sciences, Deakin University, Geelong, VIC 3216, Australia

**Keywords:** agriculture, biostimulant, soil health, productivity, tomato, rhizosphere

## Abstract

The pursuit of sustainable and productive agriculture demands the exploration of innovative approaches to improve plant productivity and soil health. The utilization of natural agricultural biostimulants, such as extracts from seaweed, fish, and humus, has gained prominence as an ecological strategy to achieve this goal. In this study we investigated the effectiveness of a fortified biostimulant extract (FBE), composed of extracts from seaweed, fish, and humus, on tomato plant physiology, productivity, and growing media properties, and estimated carbon emissions associated with tomato production. The FBE was applied to the growing media of tomato plants produced in a greenhouse, in experiments over two growing seasons. The productivity assessments demonstrated that the application of FBE significantly increased tomato fruit yield by 20% and relative marketable fruit yield by 27%, and reduced estimated greenhouse gas (GHG) emissions associated with production by 29%. FBE treatment improved plant shoot and root biomass, accelerated flower and fruit set initiation, and increased chlorophyll content in leaves, resulting in enhanced plant physiology and advanced development. FBE treatment positively influenced the availability of crucial nutrients such as nitrogen, phosphorus, and iron in the growing media. FBE promoted the growth of total active microbes in the growing media, particularly the fungal population, which plays an important role in nutrient cycling and health. These findings highlight the beneficial effects of the FBE due to enhanced plant productivity and growth, improved fertility, the promotion of beneficial plant and growing media interactions, and the reduction in estimated GHG emissions.

## 1. Introduction

The future of society relies on food that is produced in a sustainable way. Plant and soil health are the foundations for productive and sustainable agriculture. However, climate change and increasing weather extremes, such as droughts and flooding, are reducing crop yields and farmer profits [1,2]. Healthy and productive soils rely on soil ecosystem processes and the nitrogen and carbon cycles being synchronized. The intensification of agricultural practices has led to a net loss of soil organic matter which undermines crop productivity and creates a dependency for synthetic fertilizer to increase crop productivity [3,4]. These concerns have highlighted the need to extend the ‘do-no-harm’ principle of sustainable agriculture towards regenerative-based agriculture [5]. Among the principles of regenerative agriculture are (i) the reduction or elimination of the use of synthetic chemicals such as herbicides and fertilizers and (ii) establishment of beneficial plant and soil interactions, manifested by photosynthesis for plant growth and the production of root exudates, via liquid carbon pathway [5].

Combinations of natural agricultural biostimulants (such as extracts from seaweed, fish, and humus) may benefit regenerative agriculture by reconnecting natural soil processes for better agricultural productivity. The crude nature of these biostimulants makes them molecularly diverse and complex in chemical composition [6], but intriguing due to their potential synergies for sustainable agriculture.

Seaweed extracts are examples of agricultural biostimulants that farmers use to improve plant health, growth, productivity, and food quality [7,8,9]. Seaweed extracts improve plant tolerance to abiotic and biotic stresses encountered in the field, and improve the use of nutrients, such as fertilizer and water, to grow our crops [7,8,10,11]. Research has uncovered that the application of seaweed extracts to plants elicits responses in physiology, cellular signalling, gene expression, biochemical metabolites, and changes in reactive oxygen species [12,13,14,15,16,17,18]. The responses led to plant conditioning by the activation of systemic acquired resistance (SAR) and plant priming responses [16,18,19,20], and a need to sequester nutrients for plant growth. Fish protein hydrolysates are fish extracts that farmers use as agricultural biostimulants to increase crop growth, productivity, and quality [21]. Fish extracts have effects on plants such as increasing root mass, shoot length, leaf area, and chlorophyll content [21]. Fish extracts are manufactured using chemical or enzyme hydrolysis and contain organic nitrogen in the form of free amino acids and peptides [22]. Since amino acids are in a bioavailable form for rapid plant uptake and assimilation, the application of fish extracts enables plants to conserve metabolic energy that otherwise would be diverted to amino acid synthesis [21]. Humus extracts from humified organic matter, for example humic and fulvic acids, can be applied as a soil drench to improve soil structure, chelation, buffering capacity, microbial activity, and for increasing soil carbon [6]. Humus extracts promote plant and root growth when applied to plants [23]. Humic substances are extracted by liquefying humus or mineral deposits such as leonardite using chemical hydrolysis, are rich in carbon content, and contain plant available and more complex carbon forms [6]. Humic acids are complex and heterogenous in chemical structure which makes their precise characterization difficult [6]. Despite the benefits of biostimulants, there are still few reports about their effects in combination [24].

In this study, we assessed the properties of a FBE, composed of extracts from seaweed, fish, and humus. The aim of this research was to determine the effectiveness of the FBE to improve and integrate plant productivity and growing media health, as a tool suited to conventional, regenerative, and organic farming. Over two growing seasons, the FBE was applied to tomato plants potted in fertilized media, grown in a greenhouse in a replicated trial design and assessed for (i) crop productivity (total fruit yield, number of fruit, marketable fruit yield, and number of marketable fruit), (ii) plant physiology (plant root and shoot biomass, plant height, number of flower clusters, flower number, flower timing, fruit set timing, leaf chlorophyll content (SPAD value), and root and shoot nutrients), (iii) growing media and microbiology properties (total and available nutrients, total active bacteria, fungi, and yeast levels), and (iv) estimated greenhouse gas (GHG) emissions associated with the production of marketable tomato fruit.

## 2. Results

### 2.1. Plant Productivity

The tomato fruit was assessed for (i) total fruit yield, (ii) number of fruit, (iii) number of marketable fruit, and (iv) marketable fruit yield at harvest. For total fruit yield per plant, there was a significant increase by 20% or 246 g (*p* = 0.019; 95% CI (47,445)) due to the FBE treatment (Figure 1A). However, there were no significant differences observed in the total number of fruit per plant (*p* = 0.76; untreated control = 33.3 vs. FBE = 31.4) between the FBE treatment and the untreated control (Figure 1B).

For total marketable fruit yield per plant there was a significant increase of 522 g (*p* = 0.001; 95% CI (245,798)) due to the FBE treatment (Figure 1C). The percentage of marketable tomato fruit yield to the total fruit yield for the FBE treatment was 82% and for the untreated control was 55%. Consequently, the FBE treatment increased the relative marketable tomato fruit quality yield by 27% compared to the untreated control. For the number of marketable fruit per plant there was a significant increase by 38% or 3.4 fruit (*p* = 0.009; CI (1.1, 5.7)) due to the FBE treatment (Figure 1D).

### 2.2. Plant Growth

Plant physiological parameters were assessed every two weeks at 7, 21, 35, 49, 63, 77, 91, 105, and 119 days after transplant (DAT) by determining (i) the plant height, (ii) chlorophyll content, (iii) number of flower clusters, (iv) flower number, and for (v) flower initiation day and (vi) fruit set day. There was an overall significant increase (Figure 2) in the relative leaf chlorophyll content by 6.3% or 3.0 SPAD values (*p* ≤ 0.001, 95% CI (1.6, 4.4)), earlier plant flowering by 6.8 days (*p* < 0.001, 95% CI (3.7, 9.9)), and earlier plant fruit set by 6.5 days (*p* < 0.001, 95% CI (3.4, 9.6)) due to the FBE treatment relative to the untreated control. The analysis found no significant differences in plant height (*p* = 0.61; untreated control = 61.3 vs. FBE = 62.9), number of flowers per plant (*p* = 0.27; untreated control = 14.1 vs. FBE = 16.6) and number of flower clusters per plant (*p* = 0.31; untreated control = 3.89 vs. FBE = 4.10) between the FBE treatment and the untreated control.

Plant assessments of (i) shoot dry weight, (ii) root dry weight, and (iii) total plant dry weight were undertaken at the flowering (35 DAT) and harvest (119 DAT) stages. Statistical analysis found a significant increase (Figure 3) in total plant dry weight at 35 DAT by 28% or 5.3 g (*p* = 0.017, 95% CI (1.1, 9.5)) and at 119 DAT by 16.3% or 54 g (*p* = 0.018, 95% CI (11, 97)) for the FBE treatment relative to the untreated control. The FBE significantly increased the shoot dry weight at 35 DAT by 28.4% or 4.6 g (*p* = 0.025, 95% CI (0.7, 8.6)) and at 119 DAT by 14.4% or 43 g (*p* = 0.049, 95% CI (0, 85)), and increased the root dry weight at 35 DAT by 24.3% or 0.65 g (*p* < 0.001, 95% CI (0.35, 0.95)) and at 119 DAT by 38.5% or 11.6 g (*p* = 0.020, 95% CI (2.1, 21.1)) relative to the untreated control.

### 2.3. Plant Nutrient Content

Due to our interest in characterizing the flowering stage, plant nutrient content was assessed (Table 1). FBE application led to a significant increase in the plant content for most essential nutrients (N, P, K, S, Ca, Mg, Fe, Mn, Cu, Co, B), except for Zn and Mo, compared to the untreated control. The increase in the accumulation of nutrients in the plants is consistent with the FBE treatment significantly increasing biomass (Figure 3).

### 2.4. Available Nutrients in Growing Media

To assess the available nutrients at flowering, media were collected from the tomato root zone. Statistical analysis (Table 2) found the FBE treatment significantly increased the available macronutrients N and P, and Fe. However, there were no significant changes for the remaining available nutrients. In addition, the treatment had no significant effect on total N (*p* = 0.31; untreated control = 0.82% vs. FBE = 0.79%) and P (*p* = 0.29; untreated control = 1177 ppm vs. FBE = 1228 ppm) levels.

### 2.5. Microbiology in Growing Media

To assess the active microbe populations (fungi, yeasts, bacteria) at plant flowering, media were collected from the tomato root zone. Statistical analysis (Table 3) found the FBE treatment significantly increased the total active microbial population and the total active fungi, but no significant effect on the total bacterial or total yeast populations.

### 2.6. Greenhouse Gas Emissions

There was a 29% decrease in estimated GHG emissions in the greenhouse production experiments (*p* = 0.037; 95% CI (0.02, 0.70) due to the FBE treatment relative to the untreated control (Figure 4). The GHG emissions were 1.24 and 0.88 kg CO_2_-e per kg marketable tomato fruit for the untreated control and the FBE treatment, respectively. The GHG emissions value for the untreated control is similar to the average GHG emissions (1.10 kg CO_2_-e per kg) reported (using a life cycle analysis) to produce tomato fruit in an unheated greenhouse [25]. The main difference in emissions between the treatments was due to the increase in marketable yield relative to the fertilizer applied (since the greenhouse facility was not heated and no fuel was used for growing operations).

## 3. Discussion

### 3.1. The Effect of FBE on Tomato Plants

The plants treated with FBE revealed a significant increase in tomato productivity and marketable yield in a greenhouse production system with potted plants. The statistical analysis of the results found significant increases in fruit yield by 20% and the relative marketable fruit was enhanced by 27%. The increase in total yield was not accompanied by a proportional increase in the number of fruits per plant (i.e., fruit from plants treated with FBE were larger). Therefore, the use of FBE in tomato cultivation may be a practical option for achieving higher marketable yields per plant and increasing the market value of the crop.

FBE application was found to significantly improve tomato shoot biomass by 28% and root biomass by 24% at flower initiation, and increased shoot biomass by 14% and root biomass by 39% at the harvest stage. Despite the significant increases in biomass, there were no differences in plant height between the FBE-treated and untreated control plants. Other effects of the FBE treatment on plant physiology were significantly earlier flower initiation (by 6.8 days) and earlier fruit set initiation (by 6.5 days). The results are consistent with the FBE treatment (i) advancing plant development, (ii) developing a productive plant structure without excessive plant height, and (iii) increasing agronomic productivity by increasing fruit size (and not total fruit number per plant).

This study found that the FBE treatment significantly increased the relative chlorophyll content, as measured by SPAD testing, in tomato leaves across plant development. The SPAD assessment serves as a reliable indicator of photosynthetic activity, leaf nitrogen status and leaf chlorophyll content [26,27]. The sustained higher relative chlorophyll content in the leaves across the growing season is a means to efficiently capture light energy by photosynthesis for the conversion into chemical energy and consequently enhancing the plant biomass. The results are consistent with the FBE treatment influencing plant physiology. At flower initiation, the FBE treatment was found to significantly increase the accumulation of N, P, and K content in tomato plants and other essential nutrients, such as S, Ca, Mg, Fe, Mn, Cu, Co, and B. The macronutrients N, P, and K enhance plant growth and hence are frequently applied as fertilizers in modern agricultural production [28].

In this study, there is little evidence that the plant productivity and physiological responses are due to a fertilizer effect. The experiments included fertilizer in the potting soil and supplementary fertilization in the middle of the experiment (in both the untreated control and the FBE treatment). The applied fertilizer substantially exceeded the low amount of nutrients in the diluted FBE that was applied. Instead, the results may be related to the seaweed extract in the FBE since there is evidence reporting that seaweed extracts can enhance chlorophyll content and photosynthesis capacity, improve tomato growth and productivity [29], and promote early flowering and early fruit set in different crops [30,31].

### 3.2. The Effect of FBE on Growing Media

The effect of FBE treatment on media was pronounced when assessed at flowering stage of tomato plant growth. The FBE promoted significant increases in three available nutrients (N, P, and Fe), utilized by plant roots and microbes for growth, and soil mineralization cycles. For comparable results and insights, the microbiology of the media was assessed at flowering stage and revealed significant increases in the total microbial active population and specifically in the total fungal population, around the root zone. Fungi have diverse roles in crop ecology from providing nutrition for plant growth to causing plant disease [32]. For example, fungi decompose soil organic material as part of the soil carbon cycle, and the majority of terrestrial plants, including tomato plants, form root associations with arbuscular mycorrhizal fungi to increase plant uptake of P, as well as the uptake of inorganic and organic N [33,34]. At the same time, soil-borne fungi cause a range of diseases in tomato crops [35], and microbes can compete with crops for nutrients [36]. Throughout the current experiments, there were no symptoms of disease or pathogenic fungi on the tomato roots. Therefore, we hypothesize that the increase in fungal population around the roots of plants were non-pathogenic. It is important to recognize that the method used to enumerate micro-organisms in the current experiment only measured culturable organisms, which represent a small fraction of the rhizosphere. Therefore, further experiments using next-generation sequencing approaches are needed to identify and validate the rhizosphere organisms from the current study.

### 3.3. The Effect of FBE on the Plant and Soil Ecosystem

Productive agriculture relies on the connectivity of the soil and plant ecosystems. However, the natural ecosystem processes become disconnected in depleted soils that are farmed intensely with limited crop diversity. Similarly, a dependency upon synthetic fertilizer eventually results in fragmented biological, plant, and soil ecosystems [37]. A possible approach for growers to boost, connect, and regenerate the geobiochemical processes (Figure 5) is to combine the benefits provided by different biostimulant ingredients such as seaweed, fish, and humus extracts.

Seaweed extracts have been demonstrated to improve plant growth and soil health. For example, seaweed extracts have been found to increase root biomass [31,38], resulting in root systems with improved water and nutrient uptake, leading to enhanced overall plant growth and vigour [39]. The reported increase in root biomass aligns with the findings of our study, where plants treated with FBE exhibited an increase in root biomass. Furthermore, treatment with seaweed extract is reported to provide improved plant tolerance to stresses, explained by a mechanism related to plant priming [9,16].

Humic extracts (containing fulvic and humic acids) possess complex and heterogenous carbon molecules that have beneficial properties such as promoting the growth and metabolism of beneficial soil microbes [40]. Higher soil carbon content promotes fungi-dominated communities due to their superior carbon use efficiency [33]. This efficiency enables fungi to effectively utilize carbon resources, leading to increased carbon sequestration in the soil. The beneficial microbes can solubilize vital nutrients in the soil; for example, making P available to plants for uptake [41]. Humic substances have oxygen, nitrogen, and sulphur-containing functional groups that allow them to make stable complexes in the soil with metal micronutrients, such as Fe, to maintain micronutrients in their available forms [42]. Studies on grapes noted an increased uptake of Fe and P [43], and in pepper and cucumber were reported to have an increased uptake of N, P, K, Ca, and Mg, after application of humic acids in field conditions [44,45].

Fish extracts produced by hydrolysis are examples of liquid organic nitrogen extracts and contain high levels of oligopeptides, polypeptides, and free amino acids that can be absorbed by plants directly into the roots and utilized as a source of nitrogen [46]. The amino acids and peptides can complex and chelate soil micro- and macronutrients, such as Fe, so that these become more accessible to plants [47]. Additionally, amino acids and peptides can serve as source of nitrogen for soil micro-organisms, increasing microbial activity in the soil, which in turn accelerates the decomposition of organic matter, facilitating the conversion of organic nutrients into available forms [48].

In this study, we found an increase in the available N and P in the growing media with FBE treatment, while the total N and total P levels remained unchanged. The mineralization of N and P is a chemical process driven by microbial activity [49,50]. Micro-organisms can break down organic matter and release inorganic forms of N and P, making them accessible for uptake by plants [51]. The observed increase in available N and P indicates that FBE may stimulate microbes to enhance the availability of these nutrients, aligning with the literature on fungal-dominant soils [52,53]. The combination of biostimulant ingredients may enable a steady supply of nutrients and metabolites in freely available and processable forms. By emphasizing the advantages of fortifying a seaweed extract with humic acids (as available and processable carbon forms) and organic nitrogen extracts (as available and processable nitrogen forms), our study contributes to the scientific literature by providing valuable insights into optimizing soil–plant interactions and fostering sustainable and productive agricultural practices.

We focused on the flowering stage as a pivotal point that connects growing media biology, nutrition, and plant productivity. The flowering stage is a critical phase where plants have an increased demand for phosphorus, among other elements. This stage marks a crucial transition in plant development where metabolic changes occur and plants utilize nutrition from media in competition with soil microbes [54,55]. Previous research by Lu et al. [56] has indicated that micro-organisms present in the rhizosphere can influence the timing of plant flowering. Similarly, at the tomato flowering stage, Hussain, Kasinadhuni, and Arioli [38] reported a seaweed extract increased the diversity of bacterial communities linked to soil health, and increased soil available nitrogen at the root zone.

Overall, our findings highlight the effectiveness of the FBE in stimulating improvements in plant yield, fruit size, nutrient availability, microbial population in media, and root morphology. We hypothesize that the combination of these extracts strengthens the relationship between the growing media and plants, leading to enhanced plant performance, and enriches microbial populations that may facilitate improved plant nutrient utilization. A broader understanding of the effect of FBE on plant and soil biology, and their connectivity at key plant development stages, may be an approach to advance agriculture. With further validation, there may be other meaningful advantages offered by the approach of combining biostimulants since this study demonstrated a significant reduction in estimated GHG emissions by 29% associated with marketable fruit yield due to the FBE treatment.

## 4. Materials and Methods

### 4.1. Seedling Establishment

Tomato seeds (*Solanum lycopersicum* L. cv. *Grosse Lisse*) were used in the experiments (Mr. Fothergill’s Seeds, Sydney, NSW, Australia). A seed raising mix (Osmocote Seed and Cutting Premium Potting Mix) containing composted bark, sphagnum peat and coir, and controlled release fertilizer granules (N17%, P2%, K7%; Osmocote^®^, Sydney, NSW, Australia) was used to fill germination trays (80 mL volume per cell). The tomato seeds were sown one seed per cell. The seeds were then germinated by placing the trays in a temperature-controlled greenhouse for four weeks, 21 ± 3 °C, 60 ± 10% relative humidity, under natural light conditions, and the growing media were kept moist as required.

### 4.2. Experimental Plant Trials

The study was conducted in a greenhouse as two experiments over two growing seasons: spring–summer 2021 to 2022; and spring–summer 2022 to 2023 (at Bayswater, Melbourne, VIC, Australia; 37°50′36.3048″ S, 145°14′54.5136″ E). The study was laid out in a randomized complete block design with six replicates per treatment combination for season 2021–2022 and eight replicates per treatment combination for season 2022–2023. Four-week-old tomato seedlings were transplanted to individual pots (27 L) filled with Premium Potting Mix containing controlled release fertilizer (Garden Basics, Melbourne, VIC, Australia) and at flowering fertilized with Osmocote controlled release fertilizer granules (N17%, P2%, K7%; Osmocote, Sydney, NSW, Australia). A total of 24 pots for season 2021–2022 and 32 pots for season 2022–2023 containing tomato seedlings were transferred into a greenhouse exposed to local ambient spring–summer conditions.

The treatment used in the greenhouse plant experiments was a liquid FBE and the untreated control was water without FBE treatment. The FBE was chosen because of the composition of three agricultural biostimulant ingredients: seaweed extract for improving plant tolerance to stress and nutrient use efficiency, fish extract for organic nitrogen including free and bound amino acids, and humic and fulvic acids for organic carbon, all in a ratio of 6:3:1 (Trilogy^®^; Seasol International, Melbourne, VIC, Australia). The seaweed and fish extracts were made by alkaline hydrolysis from dry seaweeds (*Durvillaea potatorum* and *Ascophyllum nodosum*) and dry fish meal. The product analysis for the undiluted FBE is a Total Organic Matter content of 8.4%, carbon-to-nitrogen ratio of around 9:1 (similar to a fully digested compost), a soluble solid level set to 13% (*w*/*v*) (to standardize applications), alkaline concentrate (pH 9.9) filtered to below 150 microns, and contains 0.45% (*w*/*v*) N, 0.19% P, 2.44% K, 0.17% S, 0.44% Ca, 455 mg L^−1^ Mg, 218 mg L^−1^ Fe, 12 mg L^−1^ Zn, and 5.56 mg L^−1^ B. In addition, the FBE contains total free and bound amino acids of 2,673 ppm and fulvic acids 1.25% and humic acids 4.58%.

The plants were climate-hardened for one week before the first treatment application. The treatment of FBE was applied in the morning to the growing media around the root zone, in an application rate of 8.5 L/ha (dilution ratio of 1:250) every second week. A total of 1 L of diluted treatment solution was applied per pot, which was calculated by assessing the water-holding capacity per pot, and 1 L of water was applied for the untreated control [57]. Plants were uniformly irrigated twice daily with a timed drip irrigation system, as needed.

The plants were assessed every two weeks starting from 7 DAT, and subsequently at 21, 35, 49, 63, 77, 91, and 105 DAT until the final harvest on 119 DAT, which marked the time when 80% of tomato fruits had matured to a red colour. During the assessments, the following parameters were evaluated: plant height, chlorophyll content (SPAD), number of flower clusters, number of flowers per cluster, flower initiation day, and fruit set day.

Additional destructive assessments were carried out on plants at two developmental stages: at flowering (35 DAT), and fruit harvest (119 DAT), with six replicates per time point per treatment for season 2021–2022 and with eight replicates per time point per treatment for season 2022–2023. At 35 DAT, the assessments included: shoot and root dry weight, shoot and root nutrient concentration, available nutrient concentration in the growing media, including total nitrogen and phosphorus concentration, and media microbiology. At 119 DAT, the assessments were for: shoot and root dry weight, fruit weight per plant (total yield), number of fruit per plant, number of marketable fruit per plant, and marketable fruit weight per plant (marketable yield). All plants were pruned to a single stem according to the method described by Maboko et al. [58]. The tomato main stem was cut off below the youngest fully expanded leaf at 105 DAT to stop the plants growing up any further.

### 4.3. Plant Growth and Productivity Assessments

Plant growth was investigated for plant height [59], the number of flowers per cluster, and total number of flower clusters per plant [60]. Tomato leaf chlorophyll content was measured by using a soil plant analysis development meter, SPAD, (SPAD-502, Konica Minolta, Tokyo, Japan) [61]. The SPAD meter determines the absorbance of the leaf at two wavelength regions (red and near-infrared) and calculates a numerical SPAD value that is proportional to the amount of chlorophyll in the leaf [62]. The assessments for the flower initiation day and the fruit set day were determined by monitoring the plants daily for the appearance of the first open flower and the first fruit to set. Shoot and root dry weights were assessed by drying the plant tissue at 70 °C in a drying oven for 72 h and measured using analytical scales A&D Australasia, Melbourne, VIC, Australia) [63].

For the plant nutrient concentration analysis, the shoot (above ground plant tissue) and root of each plant was tested (by SWEP Laboratories Pty Ltd., Melbourne, VIC, Australia) using the Dumas method for total N (nitrogen) concentrations [64]. The remaining elements (P, phosphorous; K, potassium; S, sulphur; Ca, calcium; Mg, magnesium; Fe, iron; Mn, manganese; Zn, zinc; Cu, copper; Co, cobalt; B, boron; Mo, molybdenum; Se, selenium) were analysed by the methods set by the Australasian Soil and Plant Analysis Council (ASPAC) [65]. Total nutrient contents in the plant root and shoot were calculated by multiplying the concentration of each nutrient by its corresponding root and shoot biomass (dry weight) [66].

Plant productivity was assessed by measuring the fruit weight per plant (total yield), [59] and the marketable fruit weight (marketable yield) per plant; counting the number of fruit per plant [60] and the number of marketable fruit per plant. Fruits were regarded as unmarketable for processing and fresh markets, when they exhibited a green skin colour or skin diseases and fell into the very small size category (<20 g fruit weight) [67].

### 4.4. Growing Medial Nutrient and Microbiology Assessments

The growing media from the potted tomato plants were assessed at flower initiation for nutrients and total active microbial population. Samples (around 500 g) were collected from individual pots by selecting the media in contact with the roots (rhizosphere) at a depth of 10 cm. Samples were analysed (by SWEP Laboratories Pty Ltd., Melbourne, VIC, Australia) for available nutrients and total active micro-organisms (fungi, bacteria, and yeast). Nutrient analyses were conducted according to the ASPAC standardized testing methods [64]. The method of Mikhail E and Mikhail T [68] was used to test for active micro-organisms by using selective growing media and quantifying the number of colony-forming units per gram of media (cfu g^−1^) for total active fungi, bacteria, and yeast.

### 4.5. Greenhouse Gas Emissions Assessment

GHG emissions intensities were assessed for marketable tomato yields and fertilizer application rates for two tomato experiments and the emissions calculated as the carbon dioxide equivalent per kilogram of marketable tomato fruit (kg CO_2_-e per kg). The assessment for GHG emissions used an established farm-level GHG calculator [69], described for GHG emissions related to agricultural production practices across crops [70] and put into practice by the farming industry in Australia [71]. The experiment-specific GHG assessment relied on inputs during production (Scope 1) related to the consumption of electricity, fuel, and fertilizer consumption and type. In this study, the greenhouse facility was not heated and no fuel was used for growing operations, so the main driver for GHG emissions related to the fertilizer parameter (which includes estimating the nitrous oxide emissions due to nitrogen fertilizer application). It is important to note that the estimated GHG emissions in this experiment relates to greenhouse tomato production and validation in the open field has been reported using more quantitative methods [72,73].

### 4.6. Statistical Analysis

The data from both experiments (over two growing seasons) were combined for the statistical analysis. This was optimal because the analysis then used all available information on the effect of the treatments, and the treatment combinations were the same for both experiments. Linear mixed models were fitted to the data to incorporate the random effects (or blocking) structure arising from combining the experiment, and to adjust for any lack of balance arising from missing values. The random effects were specified as Season and Block nested within Season. The fixed effects were specified as Treatment (FBE vs. untreated control) and (if applicable) growth stage (35 DAT vs. 119 DAT) and its interaction with Treatment. If a set of analyses resulted in any numerical issues (such as negative estimated variance components), the random effects were simplified to Season × Block combination. If there were no missing values for an outcome, the mixed model was simplified to an analysis of variance (ANOVA). A similar ANOVA approach was applied to analyse the GHG emissions data.

Plots of residuals vs. fitted values were constructed to assess the homogeneity of variance. If the variability was shown to increase with the mean, a log transformation was applied to the data. In the case of count data (number of flower clusters or flowers per plant), a generalized linear mixed model (GLMM) with a Poisson distribution was fitted. All statistical analyses were conducted using Genstat version 22 (VSN International Ltd., Hemel Hempstead, UK). Graphs were constructed using Minitab version 19 (Minitab Inc., State College, PA, USA). The results show the overall estimated means for FBE and untreated control arising from the statistical analysis, the estimated difference between them, a 95% interval confidence for the difference (95% CI), and a *p*-value for testing the hypothesis of zero mean difference.

The analysis of some outcome variables involved additional steps. For the microbial analysis, all of the variables required a log transformation, which was performed to base 10 because of the large numbers. The means have been back-transformed to the original scale, and the estimated ratio of means is reported rather than the difference. For the plant and media nutrient analysis, variables required a log transformation, so the means have been back-transformed to the original scale, and the estimated ratio of means is reported rather than the difference.

## 5. Conclusions

This study demonstrated that the effect of an FBE, composed of seaweed, fish, and humus extracts, was to enhance tomato plant development and productivity, and nutrient availability, and a reduction in GHG emissions was calculated for a greenhouse production system with potted tomato plants. In particular, the application of FBE resulted in significant increases in tomato fruit yield and marketable yield, and was accompanied by improvements in shoot and root biomass, earlier flower and fruit set initiation, and increased chlorophyll content. Furthermore, FBE enhanced nutrient availability of N, P, and Fe. The FBE also modified populations of key microbes (fungi) in the growing media, which might explain the increase in nutrient availability. The results support the application of FBE as a valuable tool for conventional, regenerative, and organic farming practices. The FBE may offer an approach to optimize agricultural productivity while preserving the long-term fertility of soils. Further research and field trials are necessary to optimize the application of FBE in different crops and agricultural systems, and confirm the estimated reductions in GHG emissions.

## Figures and Tables

**Figure 1 plants-13-00004-f001:**
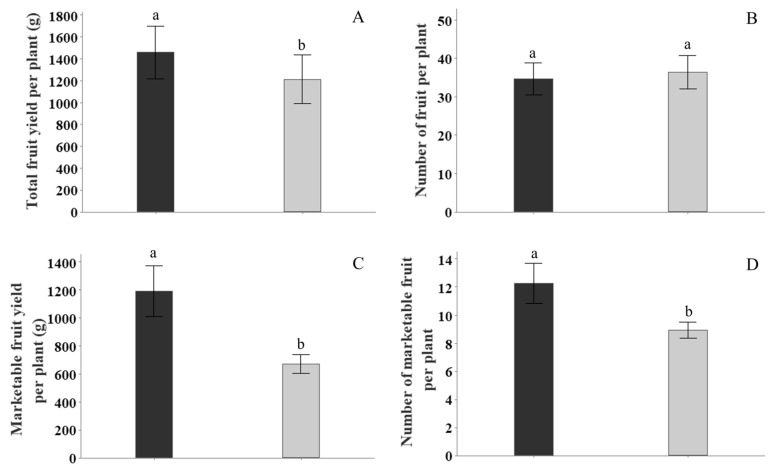
The effect of fortified biostimulant extract (FBE) on (**A**) tomato fruit weight (yield), (**B**) number of tomato fruit per plant, (**C**) marketable tomato fruit weight (yield), and (**D**) number of marketable fruit per plant, at harvest (119 DAT—days after transplant). Black columns are for the FBE treatment, and the grey columns are for the untreated controls. The FBE treatment significantly increased the total fruit yield, marketable fruit yield, and number of marketable fruit per plant relative to the untreated control. Means followed by the same letter do not significantly differ (*p* ≤ 0.05). Error bars represent SE of means (*n* = 14).

**Figure 2 plants-13-00004-f002:**
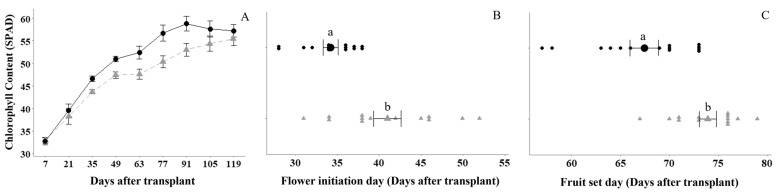
The effect of fortified biostimulant extract (FBE) on (**A**) chlorophyll content, (**B**) flower initiation day, and (**C**) fruit set day, at different DAT (days after transplant). Black circles are for the FBE treatment, and the grey triangles are for the untreated control. The means in (**B**,**C**) are shown as the larger circle or triangle. The FBE treatment significantly increased the three physiological assessments relative to the untreated control. Means followed by the same letter do not significantly differ (*p* ≤ 0.05). Error bars represent SE of means (*n* = 14).

**Figure 3 plants-13-00004-f003:**
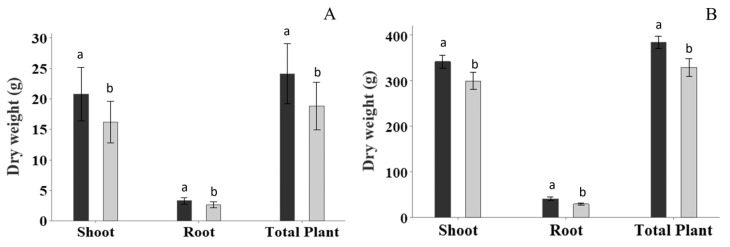
The effect of fortified biostimulant extract (FBE) on shoot dry weight, root dry weight, and total plant dry weight at (**A**) flowering (35 DAT: days after transplant) and (**B**) harvest (119 DAT) timepoints. Black columns are for the FBE treatment, and the grey columns are for the untreated control. The FBE treatment significantly increased the three biomass assessments relative to the untreated control. Means followed by the same letter do not significantly differ (*p* ≤ 0.05). Error bars represent SE of means (*n* = 14).

**Figure 4 plants-13-00004-f004:**
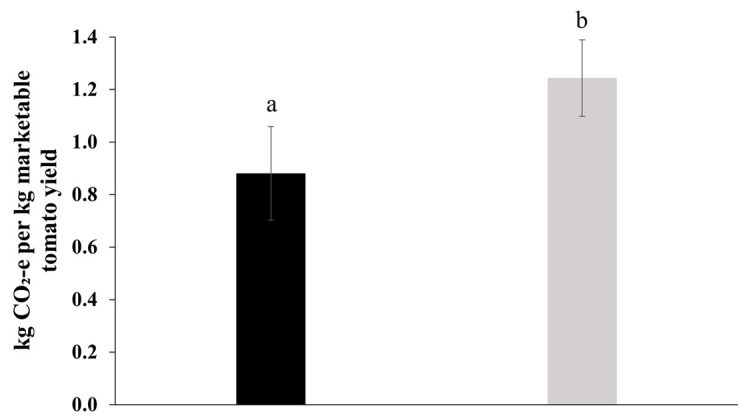
The effect of fortified biostimulant extract (FBE) on greenhouse gas emissions related to the production of marketable tomato yield. The black column is for the FBE treatment and the grey column is for the untreated control. The FBE treatment significantly decreased the greenhouse gas emissions intensity relative to the untreated control. Means followed by the same letter do not significantly differ (*p* ≤ 0.05). Error bars represent SE of means (*n* = 14).

**Figure 5 plants-13-00004-f005:**
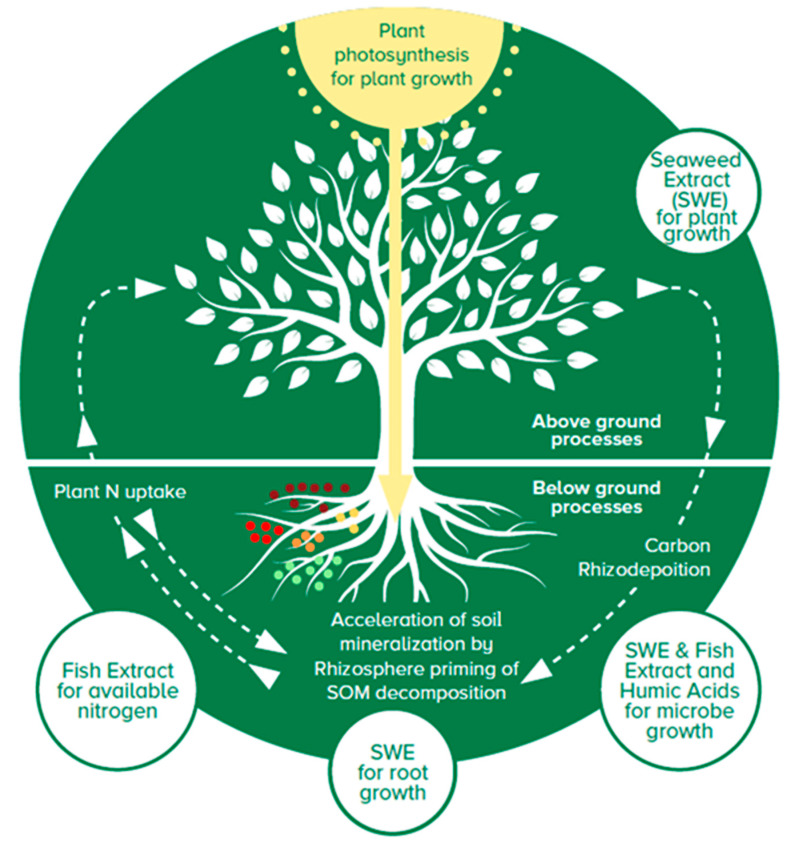
Depiction of fortified biostimulant extract (FBE) connecting and regenerating the soil ecosystem processes for better agricultural productivity and longer-term soil fertility. FBE is composed of seaweed extract (SWE), fish extract, and humic extracts. SWE provides for enhanced plant tolerance to stress and nutrient utilization. Fish extracts provide available organic nitrogen as free and bound amino acids. Humic extracts provide available carbon with humic and fulvic acids. SOM: soil organic matter.

**Table 1 plants-13-00004-t001:** The effect of fortified seaweed extract (FBE) on plant nutrient content (mg per plant) at tomato plant flowering (35 DAT: days after transplant). For the plant nutrient statistics analysis, variables required a log transformation, so the means have been back-transformed to the original scale, and the estimated ratio of means is reported rather than the difference. The 95% CI is the 95% interval confidence for the ratio. Means followed by the same letter do not significantly differ (*p* ≤ 0.05; *n* = 14).

Plant Nutrient (mg Plant^−1^)	FBE Treatment	Untreated Control	Ratio	95% CI	*p*-Value
Total N	384 a	259 b	1.48	(1.27, 1.74)	<0.001
Total P	84.6 a	67.5 b	1.25	(1.07, 1.47)	0.010
Total K	660 a	504 b	1.31	(1.10, 1.49)	0.006
Total S	168 a	126 b	1.34	(1.04, 1.72)	0.029
Total Ca	283 a	219 b	1.29	(1.07, 1.57)	0.014
Total Mg	91.7 a	68.8 b	1.33	(1.11, 1.60)	0.006
Total Fe	2.48 a	1.61 b	1.54	(1.17, 2.05)	0.006
Total Mn	1.41 a	0.97 b	1.45	(1.17, 1.80)	0.003
Total Zn	1.29 a	1.07 a	1.20	(0.87, 1.66)	0.237
Total Cu	0.39 a	0.29 b	1.34	(1.06, 1.71)	0.021
Total Co	0.0019 a	0.0013 b	1.50	(1.19, 1.89)	0.003
Total Mo	0.0026 a	0.0021 a	1.26	(0.92, 1.72)	0.141
Total B	0.49 a	0.4 b	1.22	(1.01, 1.47)	0.042

**Table 2 plants-13-00004-t002:** The effect of fortified seaweed extract (FBE) on available nutrients (ppm in growing media) at tomato plant flowering (35 DAT, days after transplant). For the nutrient statistics analysis, the difference (Diff) of the means is reported. The 95% CI is the 95% interval confidence for the difference. Means followed by the same letter do not significantly differ (*p* ≤ 0.05; *n* = 14).

Nutrients in Growing Media (ppm g^−1^)	FBE Treatment	Untreated Control	Diff.	95% CI	*p*-Value
Available N	90.1 a	48.8 b	41.3	(8.6, 74.0)	0.017
Available P	213 a	180 b	33	(10, 56)	0.009
Available K	1475 a	1421 a	54	(−85, 192)	0.420
Available S	860 a	694 a	167	(35, 368)	0.097
Available Ca	6424 a	6259 a	166	(−352, 684)	0.500
Available Mg	1245 a	1288 a	−43	(−150, 64)	0.400
Available Fe	33.5 a	25.5 b	8.0	(0.0, 16.0)	0.050
Available Mn	22.4 a	22.1 a	0.2	(−4.4, 4.8)	0.920
Available Zn	13.4 a	13.9 a	−0.5	(−1.7, 0.6)	0.330
Available Cu	17 a	18.7 b	−1.7	(−3.2, −0.1)	0.039
Available Co	0.38 a	0.45 a	0.07	(−0.19, 0.05)	0.230
Available B	0.20 a	0.15 a	0.05	(−0.00, 0.10)	0.075
Available Mo	1.53 a	1.46 a	0.07	(−0.17, 0.31)	0.550

**Table 3 plants-13-00004-t003:** The effect of fortified seaweed extract (FBE) on media microbiology at tomato plant flowering (35 DAT, days after transplant). For the microbial analysis, all the variables for colony forming units per gram of media (cfu g^−1^ media) required a log base 10 transformation and the means have been back-transformed to the original scale, and the estimated ratio of means is reported rather than the difference. The 95% CI is the 95% interval confidence for the ratio. Means followed by the same letter do not significantly differ (*p* ≤ 0.05; *n* = 14).

Media Microbiology (cfu g^−1^)	FBE Treatment	Untreated Control	Ratio	95% CI	*p*-Value
Total active population	2951 a	1905 b	1.55	(1.23, 1.95)	0.001
Total active fungi	1950 a	1132 b	1.72	(1.31, 2.27)	<0.001
Total active yeasts	0.54 a	0.39 a	1.38	(0.64, 3.01)	0.380
Total active bacteria	762 a	675 a	1.13	(0.72, 1.76)	0.560

## Data Availability

The data presented in this study are available from the corresponding author upon reasonable request.

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
