# Peer review of "Effect of a Fortified Biostimulant Extract on Tomato Plant Productivity, Physiology, and Growing Media Properties"

_plants, 2023, doi:10.3390/plants13010004_

Round 1

Reviewer 1 Report

Comments and Suggestions for Authors

It might be beneficial to indicate any statistically significant differences in the individual graphs.

Author Response

Thank you for reviewing the manuscript, and your suggestions and support for publishing the manuscript.

 Reviewer 1 Comment 1 - It might be beneficial to indicate any statistically significant differences in the individual graphs.

 Feedback 1 – We have changed the individual graphs to indicate the statistically significant differences.

Reviewer 2 Report

Comments and Suggestions for Authors

Significance analysis should be added to the Fig 1, 3, 4.

Comments on the Quality of English Language

 Minor editing of English language required.

Author Response

Thank you for reviewing the manuscript, and your suggestions and support for publishing the manuscript.

Reviewer 2 Comment 1 - Significance analysis should be added to the Fig 1, 3, 4.

Feedback 1 - We have changed the individual graphs to indicate the statistically significant differences.

Reviewer 1 Comment 2 - Minor editing of English language required.

Feedback 2 - We have reviewed the whole manuscript and made changes to improve the English language and clarify the information in the manuscript for publication.

Reviewer 3 Report

Comments and Suggestions for Authors

The manuscript titled “Effect of a fortified biostimulant extract on tomato plant productivity, physiology, and soil properties” is well written, and the results are interesting. Although the study is small, it attracts the readers and has innovative content. However, it can only be accepted for publication after making some necessary modifications.

1.       Please show data of both growing seasons. In in-vivo experiment, it is impossible that both seasons have same data.

2.       Please apply a proper post hoc test. In case of 2 variables, Student’s T-test can be applied. If authors show data of both growing seasons, ANOVA with 2 factors (Season x Treatment) can be applied, and afterwards LSD, HSD or DMR can be applied.

3.       Line 383: Please italicize “Solanum lycopersicum”. Also change in whole manuscript.

4.       Line 384-386: Please briefly write the composition of Osmocote Potting Mix.

5.       The temperature and RH can fluctuate. ±2??

6.       Why not CRD used? What was the source of variation in a greenhouse?

7.       How the FBE treatment was chosen? Were there any preliminary experiments? Present data.

8.       Authors may add more vegetative parameters based on fresh and dry weight following this paper. https://doi.org/10.1007/s00344-021-10422-2

Comments on the Quality of English Language

Minor editing of English language required

Author Response

Thank you for reviewing the manuscript, and your suggestions and support for publishing the manuscript. We have reviewed your feedback and addressed each of the points below.

 Reviewer 3 Comment 1 - Please show data of both growing seasons. In in-vivo experiment, it is impossible that both seasons have same data.

Feedback 1 -The data from both seasons were used in the analysis; they did not have the same data. Season was incorporated in the model as a random effect, with Block nested within it.  We have now replaced “Year” by “Season” in the “Statistical analysis” paragraph to make this clearer.

Reviewer 3 Comment 2 - Please apply a proper post hoc test. In case of 2 variables, Student’s T-test can be applied. If authors show data of both growing seasons, ANOVA with 2 factors (Season x Treatment) can be applied, and afterwards LSD, HSD or DMR can be applied.

Feedback 2- Season was incorporated in the model as a random effect, and not as a fixed effect, and so it is not of interest to examine the Season x Treatment interaction. The only comparison of means was between the treatment and the control, and so there are no additional post-hoc tests required. The LSD is incorporated in the 95% confidence interval.

Reviewer 3 Comment 3 - Line 383: Please italicize “Solanum lycopersicum”. Also change in whole manuscript.

Feedback 3 -We have made this change throughout the whole document.

Reviewer 3 Comment 4 -Line 384-386: Please briefly write the composition of Osmocote Potting Mix.

Feedback 4 -We have added information about the composition of the Osmocote Potting Mix.

Reviewer 3 Comment 5 - The temperature and RH can fluctuate. ±2??

Feedback 5 - We have included information about the Temp and RH variation.

Reviewer 3 Comment 6 - Why not CRD used? What was the source of variation in a greenhouse?

Feedback 6 - A CRD could have been used, but in general greenhouses show variation within them due to sun, cooling, etc., so it is usually wise to incorporate blocks in the experimental design.

By using a randomized block design, we were able to account for the variation caused by these factors by dividing the greenhouse into blocks or sections and randomly assigning treatments within each block. This design minimizes the impact of confounding variables and ensures that any differences observed between treatments are more likely to be due to the treatment itself, thereby reducing experimental error and increasing the precision of the results.

Reviewer 3 Comment 7 - How the FBE treatment was chosen? Were there any preliminary experiments? Present data.

Feedback 7 - We have clarified how the FBE treatment was chosen. We undertook one preliminary experiment for root growth phenotype however the assessment was visual and only conducted once so we have not included the results. Hence to avoid confusion, on line 419, we have deleted the sentence about a preliminary experiment.

Reviewer 3 Comment 8 - Authors may add more vegetative parameters based on fresh and dry weight following this paper. https://doi.org/10.1007/s00344-021-10422-2

Feedback 8 -Thank you for your suggestion. We have reviewed the paper. We agree that there are other useful vegetative parameters described in the paper. These assessments are practical to include in our further research.

Reviewer 3 Comment 9 - Minor editing of English language required.

Feedback 9 -We have reviewed the whole manuscript and made changes to improve the English language and clarify the information in the manuscript for publication.

Reviewer 4 Report

Comments and Suggestions for Authors

Abstract well written

Introduction well written

Line 108; no need to use short form in the legend "(FBE)" also remove from other legends

Line 113. mention values of means? how much values used for each mean,, mean replication such as SE baar replication 3? 4? 5?

In the results section, in all tables and figures, author did not use alphabet to show the statistical difference among treatment, so please use alphabet in all table and figure, easu for readers to understand the difference among treatments

Discussion, 

line 225 to 240, there is not a single suitable reference, why author not start discussion with previous published reports, he is just presenting results from start of discussion

Material and methods

line 383: (Solanum lycopersicum... it should be italic

Line 414 remove typo, correct subscript and superscript

Line 445 and 446" detialed explain the working pattern of SPAD meter. such as wavelength, and lights reference, how meter give reading, measurment mechanism of SPAD, not only describe in single lines

line 480; first time explain is enough, "Greenhouse gas emissions (GHG)"

Author Response

Thank you very much for reviewing the manuscript and providing feedback and suggestions to improve the manuscript.

Reviewer 4 Comment 1: Line 108; no need to use short form in the legend "(FBE)" also remove from other legends.

Feedback 1 – We have made this change to Figure 1 where FBE was abbreviated twice. In the remaining figures the abbreviation FBE is used repeatedly throughout the legends and requires the definition to be included for clarity.

Reviewer 4 Comment 2: Line 113. mention values of means? how much values used for each mean, mean replication such as SE bar replication 3? 4? 5?

Feedback 2 -This point has been clarified by including the number of replicates for each mean directly in the legend.

Reviewer 4 Comment 3: In the results section, in all tables and figures, author did not use alphabet to show the statistical difference among treatment, so please use alphabet in all table and figure, easy for readers to understand the difference among treatments.

Feedback 3 -We have changed the figures and tables to include alphabet letters to explain significance (in addition to the P values and 95CI information).

Reviewer 4 Comment 4: Discussion, line 225 to 240, there is not a single suitable reference, why author not start discussion with previous published reports, he is just presenting results from start of discussion

Feedback 4 -Since the section is focused on “The effect of FBE on tomato plants” we used the approach to present the results for context first since the discussion is based on the collective responses related to plant productivity and physiology.

Material and methods

Reviewer 4 Comment 5: line 383: (Solanum lycopersicum... it should be italic

Feedback 5 -We have addressed this error throughout the manuscript.

Reviewer 4 Comment 6: Line 414 remove typo, correct subscript and superscript.

Feedback 6 -We have corrected these typos for the subscript and superscript.

Reviewer 4 Comment 7: Line 445 and 446" detailed explain the working pattern of SPAD meter. such as wavelength, and lights reference, how meter give reading, measurement mechanism of SPAD, not only describe in single lines.

Feedback 7 - We have included additional wording in this section to expand the descriptions as requested.

Reviewer 4 Comment 8: line 480; first time explain is enough, "Greenhouse gas emissions (GHG)"

Feedback 8 -We have deleted the wording “Greenhouse gas emissions” as requested.

Round 2

Reviewer 3 Report

Comments and Suggestions for Authors

Dear authors, thanks for respecting my opinion and revising your manuscript accordingly. The manuscript has been improved.

Reviewer 4 Report

Comments and Suggestions for Authors

accepted